# Sacral-Nerve-Sparing Planning Strategy in Pelvic Sarcomas/Chordomas Treated with Carbon-Ion Radiotherapy

**DOI:** 10.3390/cancers16071284

**Published:** 2024-03-26

**Authors:** Ankita Nachankar, Mansure Schafasand, Eugen Hug, Giovanna Martino, Joanna Góra, Antonio Carlino, Markus Stock, Piero Fossati

**Affiliations:** 1ACMIT Gmbh, 2700 Wiener Neustadt, Austria; 2Department of Radiation Oncology, MedAustron Ion Therapy Center, 2700 Wiener Neustadt, Austria; eugen.hug@medaustron.at (E.H.); piero.fossati@medaustron.at (P.F.); 3Department of Medical Physics, MedAustron Ion Therapy Center, 2700 Wiener Neustadt, Austria; mansure.schafasand@medaustron.at (M.S.); giovanna.martino@medaustron.at (G.M.); joanna.gora@medaustron.at (J.G.); antonio.carlino@medaustron.at (A.C.); markus.stock@medaustron.at (M.S.); 4Department of Radiation Oncology, Medical University of Vienna, 1090 Wien, Austria; 5Division Medical Physics, Karl Landsteiner University of Health Sciences, 3500 Krems an der Donau, Austria; 6Division Radiation Oncology, Karl Landsteiner University of Health Sciences, 3500 Krems an der Donau, Austria

**Keywords:** radiation induced lumbosacral neuropathy, pelvic sarcoma/chordomas, dose average LET, carbon-ion radiotherapy

## Abstract

**Simple Summary:**

Late radiation-induced lumbosacral neuropathy (RILSN) is a rare, debilitating, but potentially avoidable morbidity associated with carbon-ion therapy (CIRT) for pelvic sarcomas/chordomas. This toxicity increases significantly if the long length of sacral nerves is exposed to CIRT doses > 70 Gy (RBE) [Japanese RBE model]. We propose a sacral-nerve-sparing optimized CIRT strategy (SNSo-CIRT) to minimize the risk of RILSN. This strategy is composed of (a) Contouring of individual sacral nerve roots between L5–S3 levels until sciatic nerve. (b) Restriction of doses to sacral nerves outside of high dose CTV (HD-CTV) (i.e., ‘sacral nerves-to-spare’) to doses to 5% of volume (D_RBE|LEM-I|D5%_) < 69 Gy (RBE). (c) Evaluation of robustness of SNSo-CIRT. With this strategy, doses to sacral nerves-to-spare were restricted to D_RBE|LEM-I|D5%_ < 69 Gy (RBE) for 95% of patients. Patients who developed RILSN despite the application of the sacral-nerve-sparing strategy had significantly higher D_RBE_-filtered dose-averaged linear energy transfer (LETd) on sacral nerves-to-spare. D_RBE_-filtered-LETd can be optimized along with D_RBE_ for SNSo-CIRT.

**Abstract:**

To minimize radiation-induced lumbosacral neuropathy (RILSN), we employed sacral-nerve-sparing optimized carbon-ion therapy strategy (SNSo-CIRT) in treating 35 patients with pelvic sarcomas/chordomas. Plans were optimized using Local Effect Model-I (LEM-I), prescribed D_RBE|LEM-I|D50%_ (median dose to HD-PTV) = 73.6 (70.4–76.8) Gy (RBE)/16 fractions. Sacral nerves were contoured between L5-S3 levels. D_RBE|LEM-I_ to 5% of sacral nerves-to-spare (outside HD-CTV) (D_RBE|LEM-I|D5%_) were restricted to <69 Gy (RBE). The median follow-up was 25 months (range of 2–53). Three patients (9%) developed late RILSN (≥G3) after an average period of 8 months post-CIRT. The RILSN-free survival at 2 years was 91% (CI, 81–100). With SNSo-CIRT, D_RBE|LEM-I|D5%_ for sacral nerves-to-spare = 66.9 ± 1.9 Gy (RBE), maintaining D_RBE|LEM-I_ to 98% of HD-CTV (D_RBE|LEM-I|D98%_) = 70 ± 3.6 Gy (RBE). Two-year OS and LC were 100% and 93% (CI, 84–100), respectively. LETd and D_RBE_ with modified-microdosimetric kinetic model (mMKM) were recomputed retrospectively. D_RBE|LEM-I_ and D_RBE|mMKM_ were similar, but D_RBE_-filtered-LETd was higher in sacral nerves-to-spare in patients with RILSN than those without. At D_RBE|LEM-I_ cutoff = 64 Gy (RBE), 2-year RILSN-free survival was 100% in patients with <12% of sacral nerves-to-spare voxels receiving LETd > 55 keV/µm than 75% (CI, 54–100) in those with ≥12% of voxels (*p* < 0.05). D_RBE_-filtered-LETd holds promise for the SNSo-CIRT strategy but requires longer follow-up for validation.

## 1. Introduction

The management of pelvic sarcomas/chordomas is challenging, as they are located in close proximity to several critical organs, namely the rectum, colon, urinary bladder, and sacral nerves. They often present at a locally advanced stage. Surgical resection with negative oncologic margins is the desired surgical option, and marginal/intralesional excision or gross tumor contamination of the abdominopelvic cavity should be avoided, as this may result in shorter disease-free intervals and a worse prognosis [1,2,3]. However, curative surgical resection often involves extensive resection of pelvic bones and muscles and adjacent viscera, as well as cutting through critical sacral nerves. This can result in multiple peri-operative complications like wound infection as well as permanent functional morbidities, including rectal incontinence, urinary retention, sexual dysfunction, and sensory/motor deficit [4,5]. To avoid such mutilating surgery, less invasive alternatives are explored. Chemotherapy may offer good response rates in some bone and soft tissue sarcomas; however, it requires a combination with other curative treatment options like surgery or radiotherapy to achieve adequate local control. The outcomes with radical photon-based radiotherapy in unresectable pelvic sarcomas/chordomas remain poor as curative radiation dose delivery is limited by the proximity of these tumors to several critical organs. Moreover, these tumors often display significant intra-tumoral hypoxic components, overall resulting in poor response to low LET photon-based radiotherapy [5,6,7,8,9].

Carbon-ion radiotherapy (CIRT) is one of the potentially curative treatment options for unresectable pelvic sarcomas/chordomas [10,11,12,13,14,15,16,17]. The superior biophysical advantages of carbon-ions include Bragg-Peak dose deposition, sharp penumbra, and high relative biological effectiveness (RBE) due to high linear energy transfer (LET) that translate into superior clinical outcomes in radioresistant and hypoxic sarcomas/chordomas [10,11,12,13,14,15,16,17]. Because of promising results with CIRT, there is increasing interest in offering this treatment as a functional protective alternative to surgery for operable sacral chordomas involving upper sacral vertebrae.

While we propose CIRT as a curative option and possible alternative to a morbid/mutilating surgery, emphasis should be given to the functional preservation of organs. Late radiation-induced lumbosacral neuropathy (RILSN) is an uncommon yet debilitating late toxicity of high-dose radiotherapy, also associated with CIRT for pelvic sarcomas/chordomas. RILSN is a potentially avoidable toxicity, and attempts should be made to minimize it. The risk of RILSN increases if more than 10 cm length of the nerves are exposed to doses higher than 70 Gy (RBE) [dose prescribed by Japanese RBE model: mixed beam model (MBM) or microdosimetric kinetic model (MKM)] [18]. However, the dose constraints for sacral nerves may differ with respect to the different RBE models used for CIRT dose calculations. The MKM model determines the lethal events at the micrometer scale. This model was adapted to match previous clinical CIRT experiences from NIRS/QST (National Institute of Radiological Sciences/National Institutes for Quantum Science and Technology, Chiba, Japan) with the MBM model [19,20,21,22], which was based on the LQ model for passively scattered beam, for 10% survival of Human salivary gland cells. In Japan, currently, the modified MKM (mMKM) model is used for scanning beam CIRT dose calculation. In contrast, European heavy ion facilities use the local effect model (LEM-I) for dose calculation, which is based on survival data for an idealized chordoma cell line with an alpha/beta ratio of 2 Gy [23]. Differences between these RBE models make translation of dose constraints and direct comparisons challenging. Moreover, it is unclear yet if, besides D_RBE_ statistics, several other physical entities, like high-LETd exposure to various portions of nerves, could influence the development of RILSN. To date, there is limited information available on the impact of several clinical factors, e.g., the presence of diabetes mellitus, neurodegenerative diseases, history of smoking, prior oncologic surgery, and use of chemotherapy.

In this study, we propose a strategy to restrict doses to sacral nerves without compromising high-dose target coverage. Additionally, we evaluated various clinical, D_RBE_, and LETd-related parameters that potentially influence the development of RILSN in patients with pelvic sarcomas/chordomas treated with hypofractionated CIRT.

## 2. Materials and Methods

### 2.1. Patient and Tumor Characteristics

Thirty-five patients with pelvic sarcomas/chordomas treated with CIRT at MedAustron Ion Therapy Center between August 2019 and August 2022 were evaluated. Informed consent was obtained from all the patients for anonymized data analysis and publication. Patients enrolled in an institutional prospective Registry Study (clinicaltrials.gov: NCT03049072 ethics committee: GS1-EK-4/350-2015), SNS-CIRT study (ethics committee: GS1-EK-1/208-2023) and SACRO Trial (clinicaltrials.gov: NCT0298651 ethics committee: GS1-EK-1/189-2019) were included. Adult patients with histologically confirmed non-metastatic pelvic sarcomas/chordomas between ages 24–76 years at the time of diagnosis with performance status 0–1 were included (Table 1). Patients with a prior history of radiotherapy at the same site were excluded. Nine (26%) patients underwent local oncological surgery prior to CIRT, and 2 (6%) patients received chemotherapy. Two (6%) patients had a history of diabetes mellitus. None of the patients reported pre-existing peripheral neuropathy or neurodegenerative disorder unrelated to the tumor. Five (14%) patients reported tumor associated grade 2 neuropathy at the baseline, but all the patients showed resolution of symptoms post-CIRT.

### 2.2. Treatment Simulation and Planning: Clinical

Patients were positioned in a prone position with the help of personalized MOLDCARE^®^ Cushions (polystyrene B) and non-perforated thermoplastic masks. Planning CT scans and MRI scans were acquired in the treatment position. Target volume delineation was performed as per SACRO trial protocol (ISG-2016-SACRO) and ASTRO contouring guidelines for soft tissue sarcoma [24,25]. GTV was delineated with the help of planning MRI acquired in the treatment position. MRI sequences composed of T1 post-contrast, T2 weighted, Diffusion-weighted imaging (DWI), and STIR (Short Tau Inversion Recovery) images. Pelvic sarcomas/chordomas were treated with CIRT to the LEM-I prescription doses of 73.6 (70.4–76.8) [Gy (RBE)]/16 fractions, (4 fractions/week), low dose PTV (LD-PTV) was treated with D_RBE|LEM-I|50%_ = 4.4–4.8 [Gy (RBE)] × 9 fractions followed by a sequential boost to high dose PTV (HD-PTV) with D_RBE|LEM-I|50%_ = 4.4–4.8 [Gy (RBE)] × 7 fractions. Clinical TPS RayStation 8B, 11A, and 11B SP1 (RaySearch Laboratorie AB, Stockholm, Sweden) were used for CIRT planning using multiple field optimization (MFO) based on the LEM-I model and pencil-beam dose algorithm. The treatment planning and image verification details for the CIRT plan were detailed in previous publications [26,27]. D_RBE_ was recomputed using the mMKM in a retrospective setting to evaluate adequate target coverage with both LEM-I and mMKM prescription doses [67.2 Gy (RBE)/16 fractions]. The LEM-I prescription doses and dose constraints were adapted with respect to corresponding mMKM prescription doses as per the translation schema provided by Fossati et al. [28]. The median overall treatment time was 26 days (range of 22–32). During the course of treatment, at least two re-evaluation CT scans were performed to evaluate the robustness of target and OAR dose constraints.

### 2.3. SNSo-CIRT Strategy

#### SNSo-CIRT Composed of Three Step Approach

Step-1: We contoured the individual sacral nerve roots between L5-S3 levels until they converged into the sciatic nerves. Sacral nerve roots were contoured with the help of MRI sequences [T1-post contrast, T2-weighted, +/− STIR (Short Tau Inversion Recovery) images] and adapted according to CT anatomy. Whenever the nerve roots were not visible due to the tumor, interpolation was used. The sacral nerves inside the HD-CTV are depicted in Figure 1a (magenta). The sacral nerves outside the HD-CTV are defined as ‘sacral nerves-to-spare’ (Figure 1a, green).

Step-2: CIRT plans with SNSo-CIRT were optimized to restrict D_RBE|LEM-I_ doses to 5% (D_RBE|LEM-I|D5%_, all the abbreviations are explained in Appendix A) of sacral nerves-to-spare to <69 Gy (RBE) (Figure 1b). Doses to sacral nerves-not-to-spare were restricted to (D_RBE|LEM-I_ doses to 2% volume) D_RBE|LEM-I|D2%_ < 73 Gy (RBE) (Figure 1b). In order to ascertain that no “hot spots” were inadvertently placed inside the nerves, higher priority was given to D_RBE|LEM-I|95%_ > 95% prescription doses to HD-CTV and in worst cases, D_RBE|LEM-I|D2%_ < 107% of the prescription doses (max doses) were avoided on sacral nerves inside HD-CTV.

Step 3: Most of the SNSo-CIRT plans were robustly optimized for sacral nerves against range and set-up uncertainties to ensure robust implementation of sacral-nerve-sparing in different clinical scenarios. Additionally, during treatment 2–3 re-evaluation CT scans in treatment position with immobilization were obtained, and D_RBE|LEM-I_ and D_RBE|mMKM_ doses were recomputed on these control CT scans to evaluate reliability and reproducibility of SNSo-CIRT as shown in a representative case (Figure 1c).

Additionally, cauda equina was contoured at least 5 cm beyond the LD-PTV with the help of T2-weighted MRI images. Doses to cauda equina were restricted to D_RBE|LEM-I|D5%_ < 66 Gy (RBE).

### 2.4. D_RBE_ and LETd Evaluation

Patient CT scans, structure sets, CIRT plans, CIRT doses, and DICOM files were imported into the research version of TPS RS2023B to evaluate D_RBE|LEM-I_, D_RBE|mMKM_, and LETd distribution. D_RBE|mMKM_ and LETd were recomputed and evaluated in retrospective settings. LETd in RS2023B was computed using the trichome algorithm [29]. Various D_RBE_ and LETd parameters were evaluated for whole sacral nerve roots, sacral nerves-to-spare, and cauda equina. D_RBE|LEM-I_, D_RBE|mMKM_ statistics, e.g., D_2%_, D_5%_, and max doses to different lengths (e.g., 1 cm, 2 cm, 4 cm, and 8 cm) of ‘sacral nerves-to-spare’ and ‘Whole sacral nerves’ were documented. Additionally, we hypothesized that in addition to D_RBE_, LETd could also be a contributing factor to the development of RILSN. Hence, we documented various LETd parameters: LETd_2%_ and LETd_5%_ and LETd to 1 cm, 2 cm, 4 cm, and 8 cm of the ‘sacral nerves-to-spare’ and ‘whole sacral nerves’.

LETd analysis for sacral nerves-to-spare is challenging. Outside of HD-PTV, the D_RBE_ rapidly decreases [∼6 Gy (RBE)/mm], whereas LETd may remain high. The impact of high-LETd could be negligible in areas with low D_RBE_. With this background, we decided that rather than assessing the impact of high-LETd distribution on sacral nerves, a combination of moderate to high D_RBE_ and high-LETd distribution simultaneously in the same voxel(s) or absolute volume of the nerves should be assessed as it may trigger a neuropathic event. Hence, we evaluated D_RBE_-filtered-LETd distribution on sacral nerves-to-spare and cauda equina with the help of LVH and voxel-by-voxel analysis. The stepwise approach for D_RBE_-filtered-LETd analysis was as follows:To investigate optimal D_RBE_ thresholds, we increased the threshold in steps of 10 Gy (RBE) from 0–60 Gy (RBE) and then in steps of 1 Gy from 61–73 Gy (RBE). [in other words, we excluded from the analysis the portion of the whole sacral nerves, sacral nerves-to-spare, and cauda equina that were below the mentioned D_RBE_ levels].Then, we evaluated LETd (in steps of 5–10 keV/µm from 0–200 keV/µm.) in these D_RBE_ thresholds sub-volumes with the help of LETd volume histogram (LVH) (i.e., D_RBE_ filtered LVH). The D_RBE_ thresholding resulted in a change in absolute volume in the nerve structures and made the analysis of single data points on the D_RBE_-filtered relative LVH very unreliable.Since sacral nerves are serial organs, clinically relevant damage can be triggered by injury in a very small volume. Hence, we conducted a voxel-by-voxel analysis of D_RBE_-filtered-LETd.The number of voxels (dose calculation grid size 1 mm × 1 mm × 1 mm) in each organ may vary in different patients. Hence, we extracted normalized data of D_RBE_-filtered LETd for the whole sacral nerves, sacral nerves-to-spare, and cauda equina.e.g., for D_RBE|LEM-I_ cutoff = 50 Gy (RBE), and LETd cutoff = 60 keV/µm$$Normalize\;dhigh-LETd/high\;DRBE\;voxels=\frac{No.\;of\;voxels\;in\;the\;organ\;receiving\;DRBE|LEM-I\geq 50\;Gy\;(RBE),\;and\;LETd\geq 60\;keV/µm}{Total\;No.\;of\;voxels\;in\;the\;organ}$$For each D_RBE|LEM-I_ threshold and each LETd threshold, we conducted a ROC (Receiver Operating Characteristic) analysis using a fraction of high-LETd voxels as a variable parameter. The predicted difference in the RILSN-free survival based on appropriate D_RBE_-filtered-LETd thresholds was assessed with the help of Kaplan–Meier analysis.

### 2.5. Clinical Follow-Up

Patients were followed by physical examination every 3–6 months for the first 2 years post CIRT, followed by q 6 monthly thereafter for up to 5 years. MRI imaging of the pelvis (at least T1 post-contrast, T2-weighted, STIR, Diffusion-weighted sequences) was performed at every clinical follow-up for local tumor assessment. Assessment of metastatic disease using CT-thorax and/or PET-CT was performed at 6 months post CIRT and later whenever clinically indicated. For stroma-rich mesenchymal tumors treated with radiation, the RECIST criteria have several limitations, and despite being well-defined, they are not considered the optimal tool to assess response. In our series, the detection of new solid lesions in distant organs was defined as metastatic progression; new lesions out of but in proximity to the low dose PTV (LD-PTV) were defined as regional recurrences. Local recurrences were defined as an increase in tumor volume, larger than 20% of the initial volume, as detected using volumetric slice-by-slice recontouring on follow-up MRI (considering multiple sequences), confirmed on two consecutive exams at a minimum interval of 3 months. In cases of uncertainty in the diagnosis of local or regional progression, imaging was complemented by histopathological confirmation. The treatment-associated toxicities were documented as per CTCAE v5. Various clinical components related to RILSN include neuralgia, paraesthesia, sensory/motor deficit, gait dysfunction, bladder/bowel dysfunction, sexual dysfunction, sleep, supine position tolerance, and medications were documented. Patients who developed clinical signs and symptoms of RILSN were additionally evaluated with electromyography or nerve conduction studies whenever feasible. Treatment-associated late side effects were classified as the development of new symptoms after treatment or an increase in the grade/severity of symptoms in the absence of locally progressive disease or any other pathology.

### 2.6. Statistical Analysis

RILSN-free survival was assessed using Kaplan–Meier analysis. Various LETd parameters and absolute LVHs of whole sacral nerves, sacral nerves-to-spare, and cauda equina were compared using either a (two-tailed) *t*-test for normal distribution or Mann–Whitney-U-test for non-normal distributions. Normality was tested with the Shapiro–Wilk test. *p*-value of <0.05 was considered significant (after Bonferroni correction to avoid accumulation of alpha error). The threshold for D_RBE_-filtered-LETd for sacral nerves-to-spare was evaluated using ROC analysis. The difference in the RILSN-free survival based on D_RBE_-filtered-LETd thresholds was assessed with the help of the Log-rank test.

## 3. Results

### 3.1. Clinical Outcomes of SNSo-CIRT Strategy

The median follow-up period of patients included in this study was 25 months (range of 2–53). The median age at diagnosis was 60 years (range of 24–76). The average volume of the sacral nerves contoured was similar in patients with or without RILSN. In terms of tumor characteristics, the average volume of HD-CTV and the average volume of the sacral nerves inside HD-CTV was higher in patients without RILSN than those with RILSN. Two-year overall survival, local recurrence-free survival, and progression-free survival were 100%, 93% (CI, 84–100), and 88% (CI, 77–100), respectively. Only two patients experienced a loco-regional recurrence. One case had a new nodule at the level of L5 for a tumor of S2. The area where the recurrence developed was not included in the LD-CTV, and therefore, nerve-sparing played no role in this event. The other patient had a global progression of the entire lesion suggestive of aggressive histology for which the prescribed dose was not enough to achieve local control. The nerve-sparing strategy did not compromise target coverage (HD-CTV, D_RBE|LEM-I|D98%_ > 95% of prescription dose). We consider it extremely unlikely that this event might be related to nerve-sparing as also the bulk of the tumor that was treated with 100% of the prescribed dose was progressing.

Three patients (9%) developed severe RILSN (grade ≥ 3, CTCAE v5) (Figure 2a). The average time to develop RILSN was 8 months. RILSN in two patients was confirmed using either electromyography or nerve conduction study. The third patient underwent comprehensive evaluation at a multidisciplinary orthopedic, neurology, and pain clinic, and the diagnosis of CIRT-related RILSN was established. However, an electromyography report was not available for documentation. The RILSN symptoms peaked between 6–9 months post CIRT, and the intensity of symptoms decreased in severity by 12–24 months post CIRT (Figure 2b). All patients with RILSN presented with neuralgia radiating along the course of the sciatic nerve. Two patients also reported paresthesia. One patient developed additional transient motor deficit (grade ≥ 2, CTCAE v5) and severe sacral neuralgia 4.5 months post-CIRT completion. The nerve conduction study of this patient suggested autoimmune-associated neuropathy likely due to diabetes-induced general neuropathy plus radiation-induced neuropathy for the lower limbs. This patient required hospitalization and management in the Neurology Department with intravenous immunoglobulins and methylprednisolone. Another patient with RILSN reported a history of COVID-19 vaccination a few days prior to the development of RILSN and required invasive intra-spinal injection of local anesthetics. The third patient with severe RILSN had a history of curettage surgery for the treatment of sacral chordoma prior to CIRT. All three patients were managed with oral low-dose steroids to reduce neuro-inflammation in acute phases and analgesics as per the World Health Organization (WHO) Pain Ladder (combination of NSAIDs and opioids) for long-term management. The RILSN-free survival was 91% (CI, 81–100). At the last available follow-up, RILSN grade was reduced from grade 3 to grade 2 (CTCAEv5) for all three patients.

Seven patients developed sacral-insufficiency fractures (SIF) post-CIRT, of which 3 (9%) developed severe symptoms related to SIF (grade ≥ 3, CTCAE v5), requiring in-patient management with temporary mobility restriction. All three patients received Denosumab treatment along with pain therapy. Three patients reported sleep disturbances due to local pain post-treatment. Overall, 6 (17%) patients received short-term corticosteroids with tapering doses. With strong opioids + NSAIDs + adjuvant therapy (tricyclic antidepressants, e.g., Amitriptyline or anticonvulsants like gabapentin), two (6%) patients received intrathecal injections of local anesthetics. One patient without RILSN reported a transient flare of local pain a couple of days after COVID-19 vaccination, however, he responded to symptomatic medications. One patient developed diabetic peripheral neuropathy 9 months post CIRT confirmed on electromyography. None of the patients developed bladder or bowel dysfunction as a late sequelae of CIRT.

### 3.2. D_RBE|LEM-I_ and D_RBE|mMKM_ and LETd Analysis for Sacral Nerves

By applying the SNSo-CIRT strategy, we were able to restrict average doses to sacral nerves-to-spare to D_RBE|LEM-I_ doses to 5% (D_RBE|LEM-I|D5%_) of sacral nerves-to-spare were restricted to <69 Gy (RBE) [D_RBE|LEM-I|D5%_ = 66.9 ± 1.9 Gy (RBE)] without causing significant target coverage compromise. The D_RBE|LEM-I|D98%_ > 95% of the prescription dose for HD-CTV [near minimum doses (D_RBE|LEM-I|D98%_) to HD-CTV to 70 ± 3.6 Gy (RBE), (Appendix A)]. Additionally, the robustness of the SNSo-CIRT strategy was confirmed by recomputation of doses on re-evaluation CT scans for Sacral nerves-to-spare, whole sacral nerves, and cauda equina for both D_RBE|LEM-I_, D_RBE|mMKM_ distribution (Appendix A). The D_RBE|LEM-I_, D_RBE|mMKM_, and LETd statistics for CIRT plans were not statistically different between patients with (RILSN) and those without neuropathy (No-RILSN) (Table 2). The D_RBE|LEM-I_, D_RBE|mMKM_ statistics, D_2%_, D_5%_ and doses to 1 cm, 2 cm, 4 cm, and 8 cm for the whole sacral nerves, sacral nerves-to-spare (for cauda equina doses to 0.25 cm^3^, 0.5 cm^3^, 1 cm^3^, 2 cm^3^) were not statistically different in cohort with RILSN and those without RILSN (Table 2).

LETd parameters such as LETd_2%_ and LETd_5%_ and LETd to 1 cm, 2 cm, 4 cm and 8 cm of the whole sacral nerves, sacral nerves-to-spare (for cauda equina doses to 0.25 cm^3^, 0.5 cm^3^, 1 cm^3^, 2 cm^3^, respectively) were higher for those with RILSN than those without RILSN (*p* = NS) (Table 2, Figure 3c). The relative volume-DVH and LVH comparison of sacral nerves-to-spare and whole sacral nerves were not statistically different in cohorts with RILSN and those without RILSN (Figure 3 and Appendix A). Though the LVHs for cauda equina were significantly different in those with or without RILSN, this data were available for only 21 patients (Appendix A).

### 3.3. D_RBE|LEM-I_ Filtered LETd Evaluation

LETd distribution was evaluated in sacral nerves-to-spare, whole sacral nerves, and cauda equina after filtering out low D_RBE|LEM-I_ doses, i.e., (Figure 4a,b). The voxel-by-voxel analysis revealed that most of the voxels in sacral nerves-to-spare, whole sacral nerves, and cauda equina in patients with RILSN had significantly higher LETd after filtering out for lower D_RBE|LEM-I_ doses (Figure 4c and Appendix A). On ROC analysis of normalized high-LETd/high D_RBE_ voxel data for sacral nerves-to-spare, we observed that D_RBE|LEM-I_ cutoff = 64 Gy (RBE), and LETd = 55 keV/µm (AUC = 0.7) had the highest sensitivity of predicting RILSN = 100% with specificity = 67% (Figure 4d). After filtering out for the low doses at D_RBE|LEM-I_ cutoff = 64 Gy (RBE), Kaplan–Meier analysis demonstrated that the 2-year RILSN-free survival at D_RBE|LEM-I_ cutoff = 64 Gy (RBE) in patients with <12% of voxel in sacral nerves-to-spare receiving LETd > 55 keV/µm was 100% compared those with ≥12% of voxels receiving LETd > 55 keV/µm [75% (CI, 54–100)] (*p* < 0.05) (Figure 4e). Instead, ROC analysis of normalized high-LETd/high D_RBE_ voxel data for the whole sacral nerve did not show a significant difference in RILSN-free survival. Moreover, it is practically impossible to optimize D_RBE_ and LETd for part of the sacral nerves inside HD-CTV without causing significant compromise in HD-CTV coverage. Hence, we specifically focused on sacral nerves-to-spare for D_RBE|LEM-I_ filtered LETd Evaluation.

As we can see from the AUC matrix, several D_RBE|LEM-I_ cutoffs and LETd cutoffs show AUC > 0.7. However, we wanted to select a threshold that can predict all the RILSN events, i.e., the highest safe D_RBE|LEM-I_ and LETd threshold. Moreover, we were also concerned about specificity. Considering AUC and specificity and difference in RILSN-free survival, we could select D_RBE|LEM-I_ cutoff = 64 Gy (RBE) and LETd cutoff = 55 keV/µm and threshold of voxels exceeding these values at 12% as the optimal choice. Quantitative details of this analysis are given in the Supplementary Methods. For cauda equina D_RBE|LEM-I_ cutoff = 63 Gy (RBE), and LETd = 50 keV/µm [2-year RILSN-free survival of 100% vs. 57% (CI, 30–100), Appendix A. However, for cauda equina after filtering out D_RBE|LEM-I_ less than 63 Gy (RBE), the LETd evaluation data were available for only 18 of 35 patients, and hence, D_RBE|LEM-I_ filtered LETd analysis is less relevant for cauda equina.

Apart from D_RBE_ and LETd parameters, we evaluated several clinical factors in this cohort that might influence the development of RILSN, such as history of prior local oncologic surgery, chemotherapy, and history of diabetes. However, the sample size was small for comprehensive predictive analysis.

In summary, the SNSo-CIRT strategy was able to restrict doses to sacral nerves outside of HD-CTV without causing significant compromise in target coverage. In addition to D_RBE_ optimization, additional parameters such as D_RBE|LEM-I_ filtered LETd can be explored for SNSo-CIRT plan optimization to minimize the RILSN toxicity.

## 4. Discussion

RILSN represents a notable late complication linked with high-dose radiotherapy targeting the pelvic area. Although it is a relatively uncommon toxicity, once developed, it can severely impair one or several functions and reduce quality of life. The incidence of RILSN ranges between 16–50% [18,30,31]. Pieters et al. reported 25% incidence of neuropathy at a median time of 7 years in patients treated for retroperitoneal-paraspinal sarcomas with proton and photon therapy [30]. A study from the Heidelberg Ion therapy center reported 9% incidence of peripheral nerve injury at 5 years in sacrococcygeal chordomas treated with CIRT +/− photon therapy to a dose of 66 Gy (RBE) [range 60–74 Gy (RBE), dose per fraction for CIRT (LEM-I): 3 Gy (RBE)/fraction)] [16]. Imai et al. reported 16% incidence of sciatic neuropathy at 42 months in Sacro-coccygeal chordoma patients treated with hypofractionated CIRT to a dose of 70.4 Gy RBE [dose per fraction (Japanese RBE model): 4.4 Gy (RBE)/fraction] [18]. In contrast, another publication addressing functional outcomes in pelvic bone sarcoma treated with hypofractionated CIRT reported the incidence of neurological deficit as high as 52% at 8 years [31]. With the introduction of the SNSo-CIRT strategy in our cohort, despite hypofractionated CIRT [dose per fraction (LEM-I): 4.6–4.8 Gy (RBE)/fraction], the incidence of RILSN was much lower (9%) at 2-year. However, it should be noted that the median follow-up period in the current study is shorter compared to other studies.

The pathophysiology of RILSN begins as an initial endothelial injury, triggering an inflammatory process and free radical damage to the tissue, ultimately resulting in radiation-induced fibrosis [32,33,34,35]. Histopathological examination of these nerves reveals necrosis and hyalinization of small vessels, and fibrosis and demyelination of nerves along with thickening of epineurium and perineurium [33,36]. With respect to nerve damage related to CIRT, another pathophysiologic mechanism has been reported by Imai et al. as calcium deposition in lumbosacral nerves visible on post-CIRT CT images; this can be attributed to dysfunctional remodeling of the sacral bone structure [18]. In our series, we were unable to visualize such changes. RILSN can be diagnosed clinically based on neuropathic symptoms such as neuropathic pain, paraesthesia, motor deficit, or bowel-bladder dysfunction. On electromyography, myokymic discharges are characteristic of RILSN; however, the presence of myokymic discharges and fasciculation potentials cannot rule out tumor-induced neuropathy [37]. In case of uncertain diagnosis, imaging such as MRI evaluation and PETCT can rule out disease progression or skip metastasis. RILSN picture on MRI evaluation shows diffuse hyperintensity along the nerves in T2-weighted images and/or fascicular pattern on T1 and T2-fat suppressed sequences in the absence of tumor [38,39]. In our patients, we were unable to document such hyperintense signal changes in sacral nerves. Takenaka et al. reported that neurological deficit post-CIRT was associated with the presence of large tumor volumes or those with lytic lesions [31]. However, In our study, all three patients had non-lytic sacral chordomas, and none of the patients had a tumor volume > 200 cm^3^.

One of the major contributing factors to the development of RILSN is considered high-dose radiation exposure. Data from photon therapy for pelvic malignancies suggest that the risk of RILSN significantly increases if the whole length of sacral nerves is exposed to doses up to 70–80 Gy [40]. DeLaney et al. reported no sacral neuropathy if the doses to the sacral nerve were <70 GyE in a patient treated with photon/proton radiotherapy [41]. Other studies with proton beam therapy [30,42] reported that by restricting doses to the center of cauda equina to ≤64 CGE, ambulation, bladder continence and anal sphincter continence were preserved. Lumbosacral plexus injury was also documented after a single fraction of SBRT to spine tumors with photons if doses to the nerves exceeded 24 Gy [43]. Imai et al. reported a significantly higher incidence of radiation-induced sciatic nerve injury in patients where a long length of the nerves (>10 cm) was exposed to high CIRT doses [>70 Gy (RBE)—MBM/MKM RBE model] for treatment of sacral chordomas [18]. These patients were treated with hypofractionated CIRT, with the typical dose per fraction a 4.4–4.6 Gy (RBE) with respect to the Japanese RBE model. Similarly, in our study, we treated pelvic sarcoma/chordomas with a total dose of 70.4–76.8 Gy (RBE)/16 fractions with a LEM-I prescription. To date, only a couple of publications reported the impact of high LETd and high D_RBE_ of carbon ions on OARs. Additionally, a comprehensive evaluation of the combined effect of high LETd and high D_RBE_ on OARs is lacking. Our main aim of this analysis was to evaluate the outcomes of the SNSo-CIRT strategy and identify potential factors influencing the risk of the development of RILSN. Hence, we conducted a detailed analysis of several clinical, D_RBE_, and LETd parameters on the development of RILSN in patients treated with high-dose hypofractionated pelvic CIRT.

In this study, we focused only on lower lumbosacral plexus, i.e., L5–S3 levels, as the majority of the tumors treated with CIRT involved sacrococcygeal or pelvic bone sarcoma/chordomas. We contoured individual sacral nerve roots and optimized D_RBE_ doses to sacral nerves. The contouring guidelines by Yi et al. for contouring the lumbosacral plexus are published for reference [41,44,45]. In this cohort, we used the upfront SNSo-CIRT strategy. This strategy enabled us to restrict the D_RBE|LEM-I_ and D_RBE|mMKM_ doses to the portion of the sacral nerves outside of HD-CTV. It was evident from our results that there is no difference in D_RBE|LEM-I_ and D_RBE|mMKM_ doses to whole sacral nerves, sacral nerves-to-spare, and cauda equina.

Besides the multi-model dosimetric evaluation, in the last few years, there has been growing interest in evaluating the impact of high LETd distribution of carbon ions for tumors such as chondrosarcomas, chordomas, and pancreatic tumors and non-squamous head and neck cancers [46,47,48,49,50,51]. Some of the information on LETd-guided optimization is already been published for proton beam therapy [52]. In a mixed radiation field of CIRT, LETd might not be the best representation of the LET spectra in the voxel. We could have a situation where we have 2 voxels with similar LETd values but different LET spectra [53].

The investigators from Japan evaluated the impact of high LETd on OARs. Okonogi et al. found no role of high LETd distribution or physical dose in the development of late rectal toxicity, apart from D_RBE_ in patients treated with pelvic CIRT [54]. Similarly, Mori et al. could not demonstrate any influence of high LETd on the development of sacral insufficiency fracture [55]. Nevertheless, evaluating the impact of LETd distribution on OARs to predict the risk of late toxicity is tricky, as the presence of high LETd distribution in the absence of the high D_RBE_ in the same voxel is unlikely to induce organ damage that may result in severe RILSN. The combination of high D_RBE_ and high LETd in the single voxel in OARs is potentially dangerous and could significantly reduce the D_RBE_ tolerance OARs and possibly trigger debilitating late toxicities. LETd evaluation in proton therapy reported that the concurrent effect of the RBE-weighted dose and the LETd, rather than the LETd alone, was a predictor of late side effects in the brain and ribs [56,57,58,59]. In order to study the complex effect of simultaneous overlap of high-D_RBE_ and high-LETd distribution on sacral nerves, we focused on the evaluation of LETd distribution in the region of whole sacral nerves, sacral nerves-to-spare and cauda equina, which receiving high-D_RBE_ distribution, i.e., D_RBE_-filtered-LETd evaluation. At our institution, LETd and physical dose filtration based on LET (high-LETd dose) functionalities were benchmarked and evaluated in a commercial TPS against Monte Carlo simulations [60,61].

To investigate the parameters that can be used for CIRT plan optimization in the future, rather than selecting D_RBE|LEM-I_ and LETd thresholds with maximum AUC, we selected the threshold that can predict all RILSN events at the lowest dangerous D_RBE|LEM-I_ and LETd. In our cohort, we could identify D_RBE|LEM-I_ cutoff = 64 Gy (RBE) and LETd cutoff = 55 keV/µm for nerves outside high dose CTV (the portion of the sacral nerves that can be more easily spared); we could show that avoiding exposure of more than 12% of the voxel to high LET and high dose (D_RBE|LEM-I_ > 64 Gy (RBE), and LETd > 55 keV/µm) could be protective.

The evaluation of D_RBE_-filtered-LETd for cauda equina was challenging as, in many cases, LETd alone on cauda equina was higher in the absence of high D_RBE_ in the same voxel/volume. In regard to the part of sacral nerves inside HD-CTV, little can be achieved in terms of sparing this part of the nerves without significantly compromising HD-CTV coverage as, in most cases, these nerves are in very close proximity to the gross tumor or sometimes infiltrated by the tumor. Hence, only optimal D_RBE_ and LETd thresholds for sacral nerves-to-spare seem appropriate to be included as parameters for SNSo-CIRT planning in the future.

We also evaluated different strategies for LET optimization for targets using LET optimization using ‘distal patching technique’ and ‘LETd optimization functionality’ in commercial TPS [26,27]. These studies demonstrated a positive impact of high LETd re-distribution to the center of target away from OARs. Another interesting development in the field of LET optimization (LET painting) could be the introduction of multi-ion therapy [49,50]. To minimize the late toxicity associated with CIRT, multi-ion optimization with the lighter ions such as Helium (LET of Helium is 4–40 keV/µm and ≥100 keV/µm for carbon, oxygen ions) can be used in the region where tumors are in close proximity to critical OARs, hence preventing toxicity due high LET irradiation [50]. Having said that, the use of lighter ions always has a risk of undertreating the tumor itself with lower LETd. A comprehensive validation of these beam simulations and modeling was recently published [62].

Apart from RILSN, SIF is another debilitating, painful complication observed in our cohort associated with CIRT. Development of SIF is influenced by multiple factors apart from doses to the bone, such as tumor regression kinetics, mechanical stress, and location of the tumor (e.g., proximity to sacro-iliac joint) and dose to the bone. Recent report by Mori et al. suggested that the development of SIF was directly influenced by doses to bone: D50% > 19.9 Gy (RBE), which was the only significant factor for predicting SIF [55]. Restricting doses to the bone is challenging as it depends upon the volume and location of the tumor, whereas RILSN is a potentially avoidable toxicity. If one proposes CIRT as an alternative to surgery, which may have a high risk of damaging nerve function, maximum efforts should be made to reduce debilitating toxicity like painful RILSN.

RILSN-associated symptoms can be extremely debilitating. All three patients with RILSN in our study required opioid analgesics and steroids, and two patients required additional nerve block for management; however, the symptoms resolved/decreased in intensity 6–12 months post-peak-symptoms development. Besides radiation doses, several other factors may influence RILSN, such as nerve injury during surgery and the use of neurotoxic chemotherapy (cisplatin, taxanes, vinca alkaloids) [63], history of diabetes, arthritis, smoking, and alcohol consumption. However, in our study, the clinical factors were not evaluable for predicting the risk of RILSN. Tumor regression post-CIRT could result in fibrosis around sacral nerves, resulting in nerve entrapment leading to neuropathic symptoms. We evaluated tumor regression kinetics post-CIRT in our patient cohort, which will be published in subsequent publications. However, in this cohort, we did not find any relationship between initial tumor volume or tumor regression kinetics and the development of RILSN post-CIRT.

One of the limitations of this study is the limited number of patients; hence, detailed causal analysis for the development of RILSN is difficult. CIRT is an extremely resource-intensive treatment modality. Additionally, it is an attractive treatment option for photon-resistant, unresectable/inoperable pelvic sarcomas/chordomas. Considering the tumor site and histology included in this study, i.e., pelvic sarcomas/Chordomas are also relatively uncommon and radioresistant malignancies. Given these clinical scenarios and the limited available literature on this topic, we find it reasonable to present these results from the relatively homogeneous cohort despite limited numbers.

Another limitation is that the shorter follow-up period in this study may underestimate the risk of recurrence; however, after a median follow-up period of 25 months, none of the patients developed local recurrences in the region of nerve sparing. Moreover, we acknowledge that nerve-sparing could potentially lead to an increase in local recurrences. The exact definition of local recurrences in these tumors is rather complex and not without controversies. This topic is outside the scope of this paper; however, in this patients’ group, the only two events of loco-regional progression would have been captured even with RECIST criteria, and in all the other 33 patients, there was sustained tumor shrinkage. For the purpose of the present analysis, the definition of local recurrence according to the volumetric criteria that we have selected or according to some other criteria would not lead to any significant change. Furthermore, in the SNSo-CIRT strategy, we ensured that the Dmax was avoided on the part of sacral nerves inside HD-CTV without significant under coverage of HD-CTV (D_RBE|LEM-I|D98%_ > 95% of the prescription dose, Appendix A). We also acknowledge that the RILSN can develop a few months up to 7 years post-radiotherapy. Nevertheless, our early results suggest that the SNSo-CIRT strategy can minimize the morbidity of RILSN associated with hypofractionated high-dose CIRT to pelvic malignancies. In addition, a detailed analysis of D_RBE_ and LETd parameters conducted in this study provided us with some important insights into the sacral sparing approach. Therefore, we find it reasonable to present our nerve-sparing strategies and promising outcomes that can be implemented for future patients treated with pelvic CIRT.

## 5. Conclusions

SNSo-CIRT strategy enabled us to restrict doses to sacral nerves in patients treated with CIRT for pelvic sarcomas/chordomas and limit the RILSN rate to 9% at 2 years post-CIRT. With the SNSo-CIRT strategy, there was no statistically significant difference in terms of D_RBE_ and LETd except for D_RBE_-filtered-LETd in patients with and without RILSN. In our analysis, the D_RBE_-filtered-LETd on sacral nerves outside of HD-CTV appeared as a promising parameter that can be optimized along with D_RBE_ to further minimize debilitating yet avoidable RILSN toxicity. However, these findings must be validated with long-term follow-up.

## Figures and Tables

**Figure 1 cancers-16-01284-f001:**
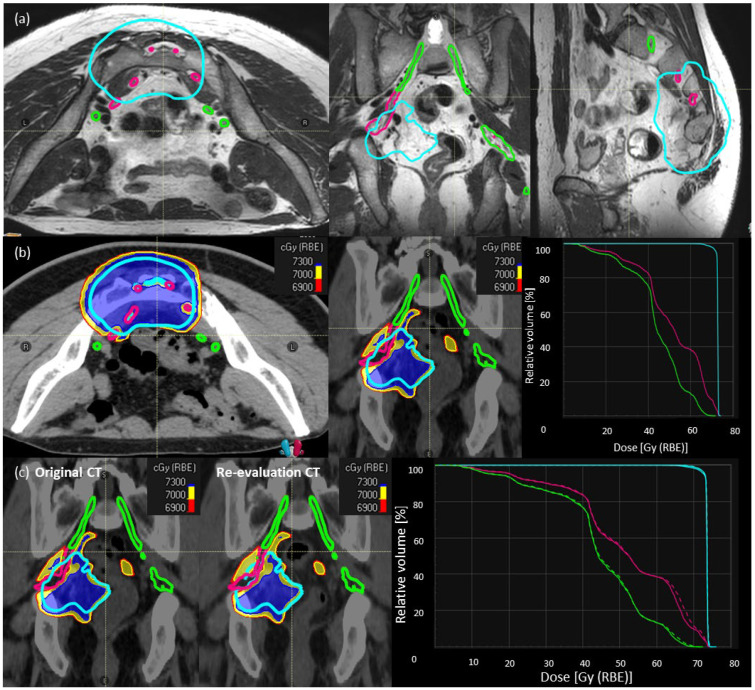
Description of SNSo-CIRT strategy. (**a**) Sacral nerves contouring in a representative case, individual sacral nerve roots (magenta) between L5–S3 levels until they converge into the sciatic nerves. The sacral nerve roots outside HD-CTV (cyan), defined as ‘sacral nerves-to-spare’ (green) (**b**) RBE weighted dose distribution [LEM-I, isodose line/surfaces red—69 Gy (RBE), yellow—70 Gy (RBE), blue—73 Gy (RBE)] showing sacral-nerve-sparing without much compromise in dose coverage for high dose CTV (cyan), doses to sacral nerves (magenta) D_RBE|LEM-I|D2%_ < 73 Gy RBE, and sacral nerves to spare (green) D_RBE|LEM-I|D5%_ < 69 Gy RBE. There are no hot spots on nerves inside high-dose CTV. (**c**) Evaluation of robustness of target and OARs dose statistics on re-evaluation CT scan compared to planning CT scan in a representative case.

**Figure 2 cancers-16-01284-f002:**
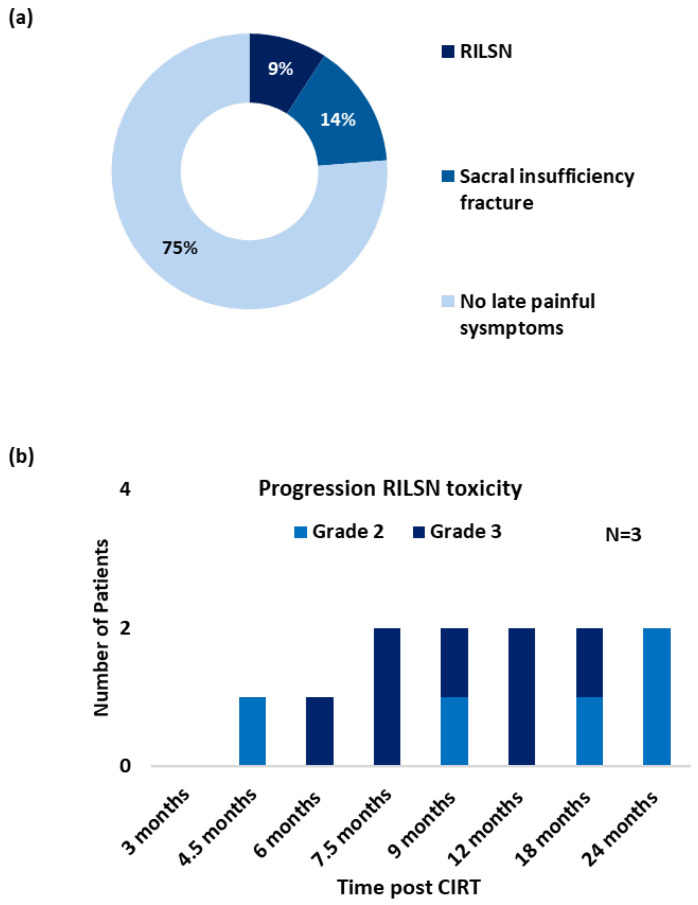
(**a**) Post CIRT late (pain) toxicity/symptom profile (grade ≥ 3, CTCAE v5), (*n* = 35). (**b**) Progression of RILSN symptoms (grade ≥ 2, CTCAE v5) in patients with RILSN (*n* = 3).

**Figure 3 cancers-16-01284-f003:**
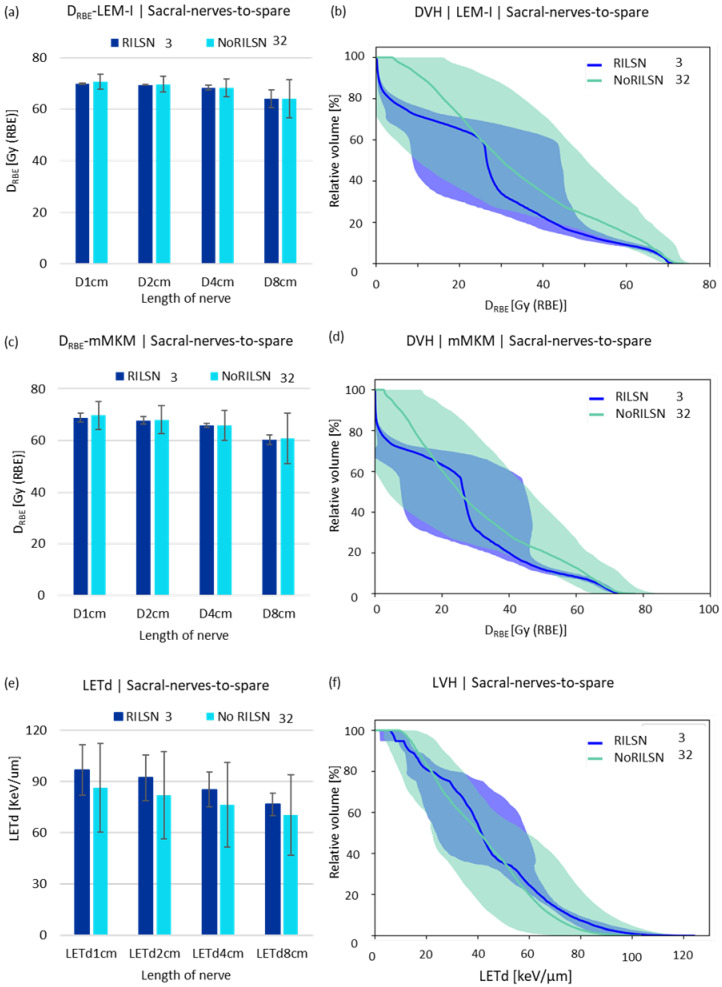
D_RBE|LEM-I_ and D_RBE|mMKM_ and LETd statistics. (**a**) LEM-I doses, (**c**) mMKM doses, and (**e**) LETd received by 1 cm, 2 cm, 4 cm, and 8 cm lengths of the sacral nerves-to-spare in patients with and without RILSN. DVH analysis of sacral nerves-to-spare in patients with and without RILSN with respect to (**b**) LEM-I and (**d**) mMKM model-based dose calculation. (**f**) Relative—volume LVH of the. Sacral nerves-to-spare of patients with RILSN and those without RILSN. (*p* = not significant).

**Figure 4 cancers-16-01284-f004:**
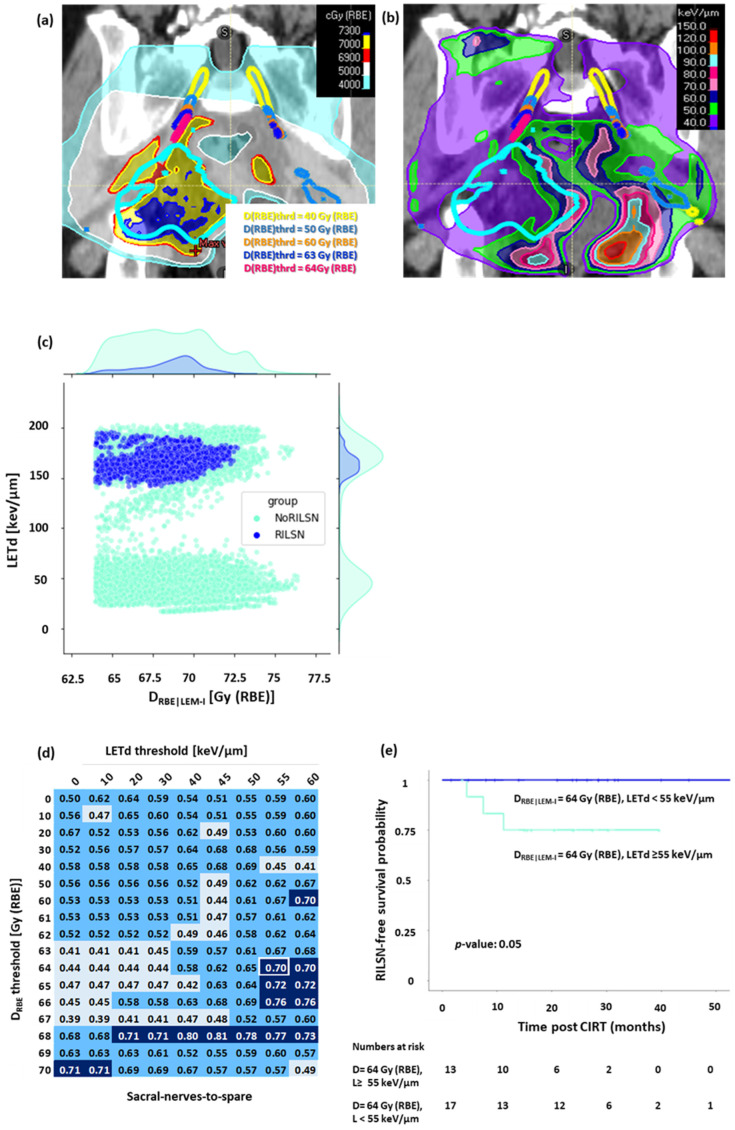
D_RBE|LEM-I_ filtered LETd evaluation for sacral nerves-to-spare, (**a**) Sub-volumes of the nerve receiving doses above the various D_RBE_ thresholds: yellow = 40. Gy (RBE), light blue: 50 Gy (RBE), orange: 60 Gy (RBE), dark blue: 63 Gy (RBE), magenta: 64 Gy (RBE). (**b**) LETd distribution on D_RBE_ threshold sub-volumes of sacral nerves-to-spare. (**c**) Voxel by voxel analysis at D_RBE_ threshold = 64 Gy (RBE) showing that most of the voxels in cases with RILSN received exposure to significantly higher LETd than those without. (**d**) AUC value matrix (light blue: AUC value < 0.5, cyan: AUC value ≥ 0.5 but <0.7, dark blue: AUC value ≥ 0.7, white frame represents AUC corresponding to D_RBE|LEM-I_ cutoff = 64 Gy (RBE) and LETd cutoff = 55 keV/µm) for D_RBE_-filtered-LETd for sacral nerves-to-spare (fraction of voxel above both D_RBE_ and LETd threshold was used as a variable parameter to create the ROC curve). (**e**) Kaplan–Meier survival analysis: at D_RBE|LEM-I_ cutoff = 64 Gy (RBE), patients with <12% voxels of sacral nerves-to-spare receiving LETd > 55 keV/µm had significantly higher 2-year RILSN-free survival (100%) than those with ≥12% of sacral nerves-to-spare voxels receiving LETd > 55 keV/µm 75% (CI, 54–100). (Note: for Figure 4c–e, n: RILSN = 3, NoRILSN = 27, D_RBE|LEM-I_ filtered LETd data for five patients without any major toxicity was not evaluable due to technical limitation).

**Table 1 cancers-16-01284-t001:** Patient Characteristics.

Patient Characteristics		No-RILSN	RILSN
		(*n* = 32)	(*n* = 3)
Age	Median (years)	61	55
	Range (years)	24–76	54–66
Gender	Male	20	2
	Female	12	1
Histology	Chordoma	25	3
	Chondrosarcoma	3	0
	Leiomyosarcoma	2	0
	Others	2	0
Surgery		8	1
Chemotherapy		1	1
Comorbidities	Diabetes	1	1
CIRT doses [LEM-I]	Median [Gy (RBE)]	73.6	73.6
	Range [Gy (RBE)]	70.4–73.6	70.4–73.6
Tumor characteristics			
Volume of HD-CTV	Mean ± SD [Gy (RBE)]	503.3 ± 402.1	127.6 ± 45.8
Volume of whole sacral nerves contoured	Mean ± SD [Gy (RBE)]	32.5 ± 4.6	30.3 ± 12.1
Volume of sacral nerves inside HD-CTV	Mean ± SD [Gy (RBE)]	6.2 ± 6.0	2.8 ± 1.0

**Table 2 cancers-16-01284-t002:** D_RBE|LEM-I_ and D_RBE|mMKM_ and LETd statistics.

D_RBE_ Statistics		No-RILSN	RILSN
		(*n* = 32)	(*n* = 3)
Whole Sacral nerve	LEM ± I		
	D_2%_ {Mean ± SD [Gy (RBE)]}	72.2 ± 2.5	71.0 ± 1.1
	D_5%_ {Mean ± SD [Gy (RBE)]}	71.5 ± 3.3	70.0 ± 1.8
	D_1 cm_ {Mean ± SD [Gy (RBE)]}	73.0 ± 3.0	72.3 ± 1.4
	D_2 cm_ {Mean ± SD [Gy (RBE)]}	72.6 ± 3.1	71.8 ± 1.1
	D_4 cm_ {Mean ± SD [Gy (RBE)]}	72.0 ± 3.3	70.6 ± 1.2
	D_8 cm_ {Mean ± SD [Gy (RBE)]}	70.7 ± 4.0	69.0 ± 2.3
	mMKM		
	D_2%_ {Mean ± SD [Gy (RBE)]}	71.5 ± 5.0	71.3 ± 1.2
	D_5%_ {Mean ± SD [Gy (RBE)]}	69.6 ± 5.2	69.9 ± 1.5
	D_1 cm_ {Mean ± SD [Gy (RBE)]}	72.7 ± 5.1	72.2 ± 1.1
	D_2 cm_ {Mean ± SD [Gy (RBE)]}	71.7 ± 5.2	71.6 ± 1.3
	D_4 cm_ {Mean ± SD [Gy (RBE)]}	70.5 ± 5.3	70.8 ± 1.4
	D_8 cm_ {Mean ± SD [Gy (RBE)]}	68.3 ± 6.0	68.8 ± 2.5
Sacral nerves-to spare	LEM-I		
	D_2%_ {Mean ± SD [Gy (RBE)]}	69.8 ± 3.0	69.3 ± 0.3
	D_5%_ {Mean ± SD [Gy (RBE)]}	67.8 ± 4.1	66.9 ± 1.9
	D_1 cm_ {Mean ± SD [Gy (RBE)]}	70.5 ± 2.9	69.9 ± 0.2
	D_2 cm_ {Mean ± SD [Gy (RBE)]}	69.7 ± 3.0	69.4 ± 0.2
	D_4 cm_ {Mean ± SD [Gy (RBE)]}	68.3 ± 3.5	68.3 ± 0.9
	D_8 cm_ {Mean ± SD [Gy (RBE)]}	64.0 ± 7.5	64 ± 3.4
	mMKM		
	D_2%_ {Mean ± SD [Gy (RBE)]}	68.4 ± 5.2	67.5 ± 1.5
	D_5%_ {Mean ± SD [Gy (RBE)]}	65.3 ± 6.2	63.7 ± 1.2
	D_1 cm_ {Mean ± SD [Gy (RBE)]}	69.7 ± 5.3	68.8 ± 1.7
	D_2 cm_ {Mean ± SD [Gy (RBE)]}	68.1 ± 5.4	67.8 ± 1.5
	D_4 cm_ {Mean ± SD [Gy (RBE)]}	65.8 ± 5.9	65.9 ± 0.8
	D_8 cm_ {Mean ± SD [Gy (RBE)]}	60.7 ± 9.8	60.1 ± 1.9
Cauda equina	LEM ± I		
	D_2%_ {Mean ± SD [Gy (RBE)]}	58.1 ± 18.3	65.3 ± 2.0
	D_5%_ {Mean ± SD [Gy (RBE)]}	54.2 ± 21.1	60.3 ± 7.8
	D_1 cm_ {Mean ± SD [Gy (RBE)]}	57.9 ± 18.2	64.6 ± 2.8
	D_2 cm_ {Mean ± SD [Gy (RBE)]}	55 ± 20.1	59.8 ± 8.6
	D_4 cm_ {Mean ± SD [Gy (RBE)]}	49.6 ± 23.0	52 ± 12.1
	D_8 cm_ {Mean ± SD [Gy (RBE)]}	39.9 ± 24.0	38.6 ± 15.6
	mMKM		
	D_2%_ {Mean ± SD [Gy (RBE)]}	52 ± 18.2	59.3 ± 3.9
	D_5%_ {Mean ± SD [Gy (RBE)]}	43.7 ± 23.5	52.8 ± 7.56
	D_1 cm_ {Mean ± SD [Gy (RBE)]}	52.0 ± 18.2	57.2 ± 3.9
	D_2 cm_ {Mean ± SD [Gy (RBE)]}	48.6 ± 19.4	51.7 ± 9.1
	D_4 cm_ {Mean ± SD [Gy (RBE)]}	43.4 ± 21.7	44 ± 12.1
	D_8 cm_ {Mean ± SD [Gy (RBE)]}	33.5 ± 22.9	32.2 ± 16.6
LETd Statistics			
Whole Sacral nerve	LETd_2%_ (Mean ± SD [KeV/µm])	81.6 ± 25.0	97.9 ± 12.6
	LETd_5%_ (Mean ± SD [KeV/µm])	73.6 ± 23.6	87.1 ± 9.3
	LETd_1 cm_ (Mean ± SD [KeV/µm])	86.2 ± 25.7	103.0 ± 11.9
	LETd_2 cm_ (Mean ± SD [KeV/µm])	81.9 ± 25.3	97.9 ± 10.9
	LETd_4 cm_ (Mean ± SD [KeV/µm])	76.3 ± 24.3	89.7 ± 7.9
	LETd_8 cm_ (Mean ± SD [KeV/µm])	70.2 ± 23.3	79.9 ± 3.7
Sacral nerves-to spare	LETd_2%_ (Mean ± SD [KeV/µm])	82.8 ± 25.2	99.2 ± 13.4
	LETd_5%_ (Mean ± SD [KeV/µm])	75.4 ± 24.2	88.4 ± 10.4
	LETd_1 cm_ (Mean ± SD [KeV/µm])	86.8 ± 25.5	103.0 ± 11.9
	LETd_2 cm_ (Mean ± SD [KeV/µm])	81.9 ± 25.3	97.8 ± 10.8
	LETd_4 cm_ (Mean ± SD [KeV/µm])	76.0 ± 24.3	89.6 ± 8.0
	LETd_8 cm_ (Mean ± SD [KeV/µm])	69.4 ± 23.4	79.8 ± 3.8
Cauda equina	LETd_2%_ (Mean ± SD [KeV/µm])	111.6 ± 74.3	173.0 ± 6.7
	LETd_5%_ (Mean ± SD [KeV/µm])	104.9 ± 72.1	162.7 ± 13.9
	LETd_1 cm_ (Mean ± SD [KeV/µm])	106.9 ± 75.7	168.8 ± 13.3
	LETd_2 cm_ (Mean ± SD [KeV/µm])	102.2 ± 74.1	158.2 ± 20.7
	LETd_4 cm_ (Mean ± SD [KeV/µm])	87.2 ± 72.1	124.9 ± 45.3
	LETd_8 cm_ (Mean ± SD [KeV/µm])	72.2 ± 67.3	89.8 ± 61.9

## Data Availability

The data presented in the current study are available from the corresponding author (A.N.) upon reasonable request.

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
