# Peer review of "Sacral-Nerve-Sparing Planning Strategy in Pelvic Sarcomas/Chordomas Treated with Carbon-Ion Radiotherapy"

_cancers, 2024, doi:10.3390/cancers16071284_

Round 1

Reviewer 1 Report

Comments and Suggestions for Authors

The Authors proposed a sacral-nerve-sparing optimized carbon-ion radiotherapy strategy to minimize risk of radiation-induced lumbo-sacral neuropathy.

The topic is extremely interesting and the study well designed.

The paper is well prepared.

Although the series is small, it is homogeneous.

I have a few minor concerns.

Was tumor neuropathy solved before radiotherapy?If not, how could this be excluded from side effects of RT? Please discuss.

Please detail further on which MRI sequences were used for nerve identification.

Extremely short follow up for some patients. This may underestimate the risk of recurrence. Please acknowledge as a limitation.

Also, it might be interesting to verify where LR occurred. Did it developed in the "nerve sparing dose" area?

Author Response

We sincerely thank Reviewer #1 for reviewing our work and providing constructive feedback. We tried to address all the issues raised by Reviewer #1 in following format and the changes made in the manuscript are highlighted in red. Wherever relevant we highlighted original text in green to bring to the attention of reviewers.

Detailed response to all the comments can be found in attached document

Reviewer 2 Report

Comments and Suggestions for Authors

One of the limitations of this study is the limited number of patients and shorter follow up period as RILSN can develop at few months upto several years post radiotherapy.

Another limitation of this study is that due to small number of patients detailed causal analysis for development of RILSN is difficult.

However, our early results suggest that the SNSo-CIRT strategy can minimize the morbidity of RILSN associated with hypofractionated high dose CIRT to pelvic malignancies.

In addition a detailed analysis of DRBE and LETd parameters conducted in this study provided us some important insights on sacral sparing approach for future patients treated with pelvic CIRT.

SNSo-CIRT strategy enabled us to restrict doses to sacral nerves in patients treated with CIRT for pelvic sarcomas/chordomas and limit RILSN rate to 9% CIRT.

With SNSoCIRT strategy there was no statistically significant difference in terms of DRBE and LETd except for DRBE-filtered-LETd in patients with and without RILSN. Our analysis suggests that DRBE-filtered-LETd as a promising parameter along with DRBE could further minimize debilitating yet avoidable RILSN toxicity.

Author Response

We sincerely thank Reviewer #2 for reviewing our work and providing constructive feedback. We tried to address all the issues raised by Reviewer #2 in following format and the changes made in the manuscript are highlighted in red. Wherever relevant we highlighted original text in green to bring to the attention of reviewers.

Detailed responses to the reviewers comments can be found in attached documents.

Reviewer 3 Report

Comments and Suggestions for Authors

This is a nice, complex paper. 

1. Chordoma is well treated by protons and perhaps even photons - this is only about carbon and as such makes it of very, very limited impact. There has been an assumption that carbon is better - but the DKRZ prostate trial shows that that is not necessarily the case. As such, there is a huge, huge, huge gaping hole in this paper because it fails to look at protons. It needs to do so to have the proper impact….protons could actually be less toxic by a lot. 

2. The paper has so many acronyms that it is almost unreadable. The abstract is horrendous. I looked at it as a new reviewer in the email from the editors and I was so annoyed by the thing that I almost refused to review this. You MUST make it more readable in terms of acronyms. I don’t know how you should do that, but in its current state it is not something anyone will actually read. It is far too much work to read as is. This must be fixed before publication. Or it should not be published. 

3. I’d like a simple table with LET and complication presented with sparing and non sparing as broken out groups….but something making the connection there more visual. I find the DVH’s too complex for this to be communicated.

Author Response

We sincerely thank Reviewer #3 for reviewing our work and providing constructive feedback. We tried to address all the issues raised by Reviewer #3 in following format and the changes made in the manuscript are highlighted in red. Wherever relevant we highlighted original text in green to bring to the attention of reviewers.

Detailed response to reviewers comments can be found in attached document.

Reviewer 4 Report

Comments and Suggestions for Authors

This study focuses on minimizing radiation-induced lumbo-sacral neuropathy (RILSN) using a sacral-nerve sparing optimized carbon-ion therapy strategy (SNSo-CIRT) in 35 patients with pelvic sarcomas/chordomas. The patients were treated at the MedAustron Ion Therapy Center between August 2019 and August 2022. The key methodological steps included the contouring of individual sacral nerve roots from L5 to S3 levels, optimizing CIRT plans to restrict doses to sacral-nerves-to-spare (DRBE|LEM-I|D5% < 69 Gy RBE), and evaluating the robustness of these optimized CIRT treatments. The results showed that three patients (9%) developed late RILSN (≥G3) post-treatment, highlighting a significant reduction in the incidence of this debilitating condition. The median follow-up was 25 months, demonstrating a RILSN-free survival rate at 2 years of 91%. The study also observed that sacral-nerves-to-spare in patients with RILSN had significantly higher DRBE-filtered dose-averaged linear energy transfer (LETd), suggesting that DRBE-filtered-LETd could be a promising parameter for further optimizing SNSo-CIRT strategy, albeit needing confirmation with longer follow-up.

General Comments:

1.       The findings of the study are original and tackle a question of high relevance within the field of radiation oncology. The focus on minimizing RILSN using sacral-nerve sparing strategies is a significant contribution. The technical depth of the paper, however, suggests it may cater more effectively to medical physicists rather than a broader audience of oncologists, including radiation oncologists, due to its specialized nature.

2.       This research is a pooled analysis of patients from two different trials, showcasing a comprehensive approach to patient selection and treatment assessment. Although no preliminary sample size calculation is mentioned, which is typical for studies on rare tumors, the Institutional Review Board approval and clinical trial registration are duly noted, underscoring the study ethical standards.

3.       The manuscript provides an exceptionally detailed description of the techniques employed, the analytical methods, and the results. However, the absence of adherence to established guidelines like CONSORT in reporting results is noted.

4.       The conclusions are well-supported by the data presented, even if the follow-up period is relatively short for a complete assessment of late toxicity. This limitation is acknowledged, which is prudent given the nature of the study and the outcomes being measured.

5.       The study small sample size is identified as a weakness, although it is understandable given the rarity of the tumors being studied. The short follow-up period is also noted, limiting the reliability of data on late toxicity.

6.       The paper is well-organized and clearly written, making it accessible despite its complex subject. However, the specialized focus may present challenges to readers outside the field of medical physics.

7.       The quality of figures and clarity of tables are highlighted as strengths, contributing to the paper overall effectiveness in conveying key findings.

Specific Comments

1.       Abstract - The notation "DRBE|LEM-I|D50% = 73.6 (70.4-76.8) Gy (RBE)/16 fractions" requires clarification for readers unfamiliar with the specific terminology and concepts of carbon-ion radiotherapy (CIRT).

2.       Abstract - The statement "With SNSo-CIRT, sacral-nerves-to-spare, DRBE|LEM-I|D5% = 66.9±1.9 Gy (RBE) and HD-CTV, DRBE|LEM-I|D98% = 73.2±2.7 Gy (RBE)" introduces critical data but lacks context regarding the significance of these metrics for the uninitiated reader.

3.       Introduction - The passage "(RBE) [dose prescribed by non-LEM-I" utilizes abbreviations that are relevant to the study context but might not be universally understood.

4.       Results - The methodology for evaluating local recurrence-free survival is mentioned but lacks detailed explanation regarding criteria for defining local recurrence, the imaging modalities used, the timing of evaluations, and the specific locations of any recurrences relative to the spared nerve volumes.

5.       Discussion - The sentence "RILSN a significant late toxicity associated with high dose radiotherapy to pelvic region." is missing a verb, affecting its clarity.

Author Response

We sincerely thank Reviewer #4 for reviewing our work and providing constructive feedback. We tried to address all the issues raised by Reviewer #4 in following format and the changes made in the manuscript are highlighted in red. Wherever relevant we highlighted original text in green to bring to the attention of reviewers.

The detailed response can be found in attached file.
